# Loss Landscape Diagnosis for Gradient-Based Gray-Scott System Inversion: Disentangling the Roles of PINN Components

## Abstract

Gradient-based inversion of reaction-diffusion systems is typically approached via surrogate models or physics-informed neural networks (PINNs), while the most direct route, backpropagation through the PDE's structure itself, has largely been avoided. We pursue this direct route as a diagnostic probe, backpropagating a steady-state loss through unrolled Gray-Scott simulation to recover its parameters, with no surrogate or neural-network augmentation. Optimization fails to converge, and plotting the landscape directly locates the failure in its geometry—flat plateaus with no gradient signal, bounded by sharp cliffs that align with bifurcation boundaries—a structure that recurs across loss functions and is inherited however the gradients are routed to parameters. Reading this minimal setup as an ablation of PINN, we disentangle each component's role: with the neural network fixed, the residual loss is quadratic in the PDE parameters and yields a smooth landscape, so it alone already avoids the pathology, by implicitly encoding the full PDE dynamics across all initial conditions. The neural network, for its part, cannot repair an ill-posed parameter subspace, and so serves only to complete the observed data—a division of labor not previously made explicit. These findings carry concrete design implications for PINN-type methods and a broader heuristic on when added dimensions actually help.

## 1 Introduction

Inverse parameter estimation that recovers the governing parameters of a dynamical system from observed outputs arises across scientific domains, from developmental biology (Kondo, 2022) to computational neuroscience (Lefèvre & Mangin, 2010). Reaction-diffusion systems are a canonical class of such problems: their parameters determine qualitatively distinct pattern-forming behaviors, and inferring them from steady-state or non-terminal observations has direct relevance to both physical modeling and biological pattern analysis (Kondo, 2022).

Direct backpropagation is the most fundamental optimization mechanism in machine learning, more efficient in terms of information flow than indirect approaches such as evolutionary strategies, surrogate-based methods, or reinforcement learning. However, practical applications of machine learning to dynamical system inversion in physics have largely avoided this direct route, favoring surrogate models (Schnörr & Schnörr, 2023) or neural-network-augmented approaches such as PINNs (Raissi et al., 2019). We assume this is because the parameter-to-solution map of nonlinear reaction-diffusion systems is expected to involve irregularity. However, the specific challenges faced by direct gradient-based methods in this setting—how pathological the loss landscape[1] is, whether it can be transformed into a well-behaved equivalence, whether backpropagation makes any progress, and whether existing approaches target the actual obstacles—have not been systematically investigated. To answer these questions, we pursue this route in a minimalist form: backpropagation through

---

[1]The term *loss landscape* in deep learning typically refers to the high-dimensional parameter space of neural networks, analyzed with tools such as Hessian eigenspectra or random-direction projections. We use it more broadly, for the geometry of the loss over whatever parameter space is under investigation—here either the low-dimensional PDE parameter subspace or the joint space that also includes the neural network parameters.

unrolled simulation steps, without surrogate approximations or neural network augmentations. We choose the grid-searchable four-parameter Gray-Scott system as a fully inspectable testbed to investigate the behavior and outcomes of gradient-based optimization, aiming for conclusions that extend to broader reaction-diffusion and PDE inverse problems.

Empirical results reveal flat plateaus with negligible gradient signal and sharp cliffs that obstruct navigation in the loss landscape, and that backpropagation makes no reliable progress without deliberate intervention. Read as an ablation of PINN, these results led us to disentangle the roles of its components: the residual loss yields a well-behaved landscape by encoding full PDE dynamics—unlike the limited parameter-to-solution map probed here—while the neural network is unable to improve the parameter landscape, only serving to fill in the missing observations. These findings carry direct design implications for PINN and broader neural-network-enhanced inverse approaches.

## 2 Diagnostic Setup

### 2.1 Forward Model and Backpropagation

The Gray-Scott reaction-diffusion system models the spatiotemporal evolution of two chemical species $u$ and $v$ through the following equations (Delgado et al., 2017; Gandy & Nelson, 2022):

$$\frac{\partial u}{\partial t} = D_u \Delta u - uv^2 + F(1 - u) \tag{1}$$

$$\frac{\partial v}{\partial t} = D_v \Delta v + uv^2 - (F + k)\, v, \tag{2}$$

where $D_u$ and $D_v$ are diffusion coefficients, $F$ is the feed rate, and $k$ is the kill rate. Depending on these parameters, the system exhibits qualitatively distinct steady-state patterns, ranging from uniform solutions to spatially structured patterns such as spots, stripes, and labyrinthine structures (Kondo, 2022). The sensitivity of steady-state pattern type to parameter values is central to the inverse problem we study.

The system we investigate is set to back propagate a loss through the intact structure of a time-unrolled stepping algorithm of the Gray-Scott model. With the time step and spatial grid spacing both set to unity ($\Delta t = \Delta x = \Delta y = 1.0$), each step follows:

$$u_{next} = u + D_u \Delta u - uv^2 + F(1 - u) \tag{3}$$

$$v_{next} = v + D_v \Delta v + uv^2 - (F + k)\, v, \tag{4}$$

loss defined on how $v$ deviates from targets.

We use this backpropagation-based optimization to fit $D_u$, $D_v$, $F$, and $k$ for the distribution of a batch of 512 similar target patterns sized $128 \times 128$, with variations coming from the limited randomness in their initial conditions (IC; see Appendix A). Backpropagation is done through all unrolled steps without truncation, as computational resource was sufficient under the non-truncated setting, and no gradient explosion/vanishing was observed.

**Target patterns.** Target patterns are generated using the same time stepping functions, boundary conditions, and initial conditions as training, fixing parameters as $D_u = 0.16$, $D_v = 0.08$, $F = 0.035$, and $k = 0.065$, until no pixel differs from the last step more than $10^{-4}$ or until we reach the maximum step 50000.

**Optimization objective.** We calculate the 2D power spectrum of the generated pattern and a sampled target pattern using

$$S = \log\big(\text{fftshift}\big(\,|\mathcal{F}(\tilde{v})|^2 + \epsilon\big)\big) \in \mathbb{R}^{H \times W}, \tag{5}$$

where

$$\tilde{v} = v - \frac{1}{HW} \sum_{i,j} v_{i,j}. \tag{6}$$

Then, we use L2 loss between $S_{target}$ and $S_{generated}$.

We also use an "windowed" 2D power spectrum, which calculates the same 2D power spectrum, but in each of the 64 $16 \times 16$ windows of a $128 \times 128$ image. The 64 spectrum windows for each image are then reassembled into a data matrix of the original image size ($128 \times 128$). A windowed loss then equals the L2 loss applied to the reassembled matrices, between target and generated.

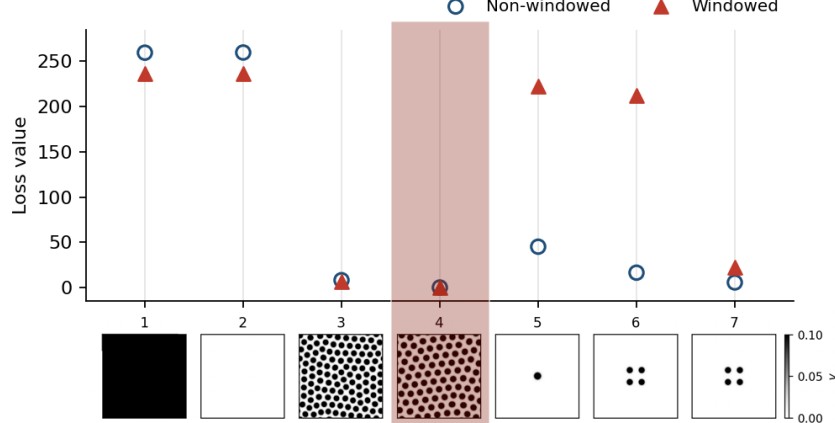

Figure 1: Sampled steady-state patterns and their non-windowed and windowed 2D power spectrum losses. Each x-axis panel shows the corresponding steady-state pattern.[2] **#1:** from initial parameters. **#2–3:** pivoting points (after 8918 and a further 98 iterations; see Appendix B). **#4:** ground truth (highlighted). **#5–7:** representative training samples.

## 2.2 Safeguards and Parameter Constraints

1. Our adaptive learning rate shrinks the learning rate until the next parameters does not generate NaN, Inf, or non-$[0, 1]$ for $v$, completely avoiding these values.

2. We restrict ranges of parameters $D_u$, $D_v$, $F$, $k$, by setting them as softplus($log\_D_u$), softplus($log\_D_v$), $0.1 \cdot$ sigmoid($raw\_k$), and $0.1 \cdot$ sigmoid($raw\_F$), adding another layer of parameters $log\_D_u$, $log\_D_v$, $raw\_k$, $raw\_F$ at the entry of the network.

3. The empirical range of $D_u$ and $D_v$ satisfies the 2-dimensional CFL criterion: $D\Delta t \left( \frac{1}{\Delta x^2} + \frac{1}{\Delta y^2} \right) \leq \frac{1}{2}$. With $\Delta x = 1.0$, $\Delta y = 1.0$, and $\Delta t = 1.0$, this means we have $D \leq \frac{1}{4}$ in most cases, although we do not take dedicated measures to strictly ensure it.

4. The initial state generation mechanism (Equations 9 and 10) is shared between target and training, so that our training goals are maximally simplified to focus on finding the four parameters $D_u$, $D_v$, $F$, and $k$.

## 3 Training Results

Gradient-based optimization fails to converge under this setup, for reasons that turn out to lie in the loss landscape rather than in any particular loss function. We establish this in the two subsections blow.

---

[2]The colormap is clipped to $[0.0, 0.1]$ to enhance contrast, as the $v$ component of the Gray-Scott model concentrates in this range under the tested parameter regimes.

### 3.1 Loss Values Carry No Convergence Signal

Across training, the loss stays within a narrow high band of 245.0–270.0 with no downward trend, departing from it only as sparse, isolated drops. In every such excursion that we allowed to continue, the loss climbed back to the high band within a few iterations.

A low loss, moreover, does not reliably indicate a correct fit. Among the sampled configurations in Figure 1, two reach comparably low loss, differing only marginally, but correspond to qualitatively different outcomes: one matches the target steady-state pattern (#3) while another does not (#7).

The matching configuration (#3) was found incidentally: we interrupted training at the moment its loss dropped, and only post-hoc inspection revealed that it matched the target. Because the run was cut off there, we did not observe whether its loss would have climbed back as every uninterrupted excursion did; we expect it would have. The parameter trajectory reaching this configuration is recorded in Appendix B.

Taken together, these observations show that the loss provides no usable signal for convergence: the low-loss configurations we encountered were reached incidentally rather than by descent, and could not be distinguished from spurious low-loss configurations by their loss value alone.

### 3.2 The Problem Persists Across Loss Functions

A natural hypothesis is that the loss function itself is at fault: the 2D power spectrum loss assigns low values to non-target patterns (Figure 1), which could both explain the spurious low-loss excursions and suggest an easy fix. To test this, we replaced it with the windowed 2D power spectrum loss (described under "optimization objective" in Section 2.1), which enforces that the dominant Fourier frequencies come equally from all sub-regions of the 128×128 field.

The windowed loss behaves no better. It produces the same trendless fluctuation, now centered around 230, with no converging trend. Evaluated on the seven parameter sets of Figure 1, its values remain polarized: each pattern sits either marginally above the target loss or up in the uniform-region range, with little in between. The windowed loss does improve separability for some non-matching patterns relative to the non-windowed version—compare the two series in Figure 1—but this sharper discrimination at isolated points does not translate into a usable gradient between them.

We can rule out insufficient exploration as the cause. Step sizes were small enough that meaningful parameter change required hundreds of iterations, and training ran for tens of hours; the stagnation is not an artifact of too-short or too-coarse a search. Because two structurally different loss functions produce the same polarized, trendless behavior, the obstruction is unlikely to reside in the loss function. This points instead to the geometry of the loss landscape itself, which we examine directly in Section 4.

## 4 Further Probing the Loss Landscapes

We now investigate possible issues in the loss landscape, regardless of loss functions. To further understand the shapes of the loss landscapes, we plotted some cross-sections of the landscape.

### 4.1 Cross-Sections: Single Parameters and Pairs of Parameters

We show the one-dimensional cross-sections along $k$, $F$, $D_u$, and $D_v$ in Figure 2. Along each of these parameter variation directions, we also record a sequence of animations sweeping that direction. The animation file links are in Appendix D.

We show loss values in two-dimensional planes formed by varying pairs of parameters in Figure 3. Two such planes are presented, one for $F$-$D_v$ and the other for $F$-$k$. Figure 4 shows steady-state patterns arranged in a $7 \times 7$ matrix that illustrate the same $F$-$k$ region, providing a more intuitive view of the trend underlying the $F$-$k$ loss plot.

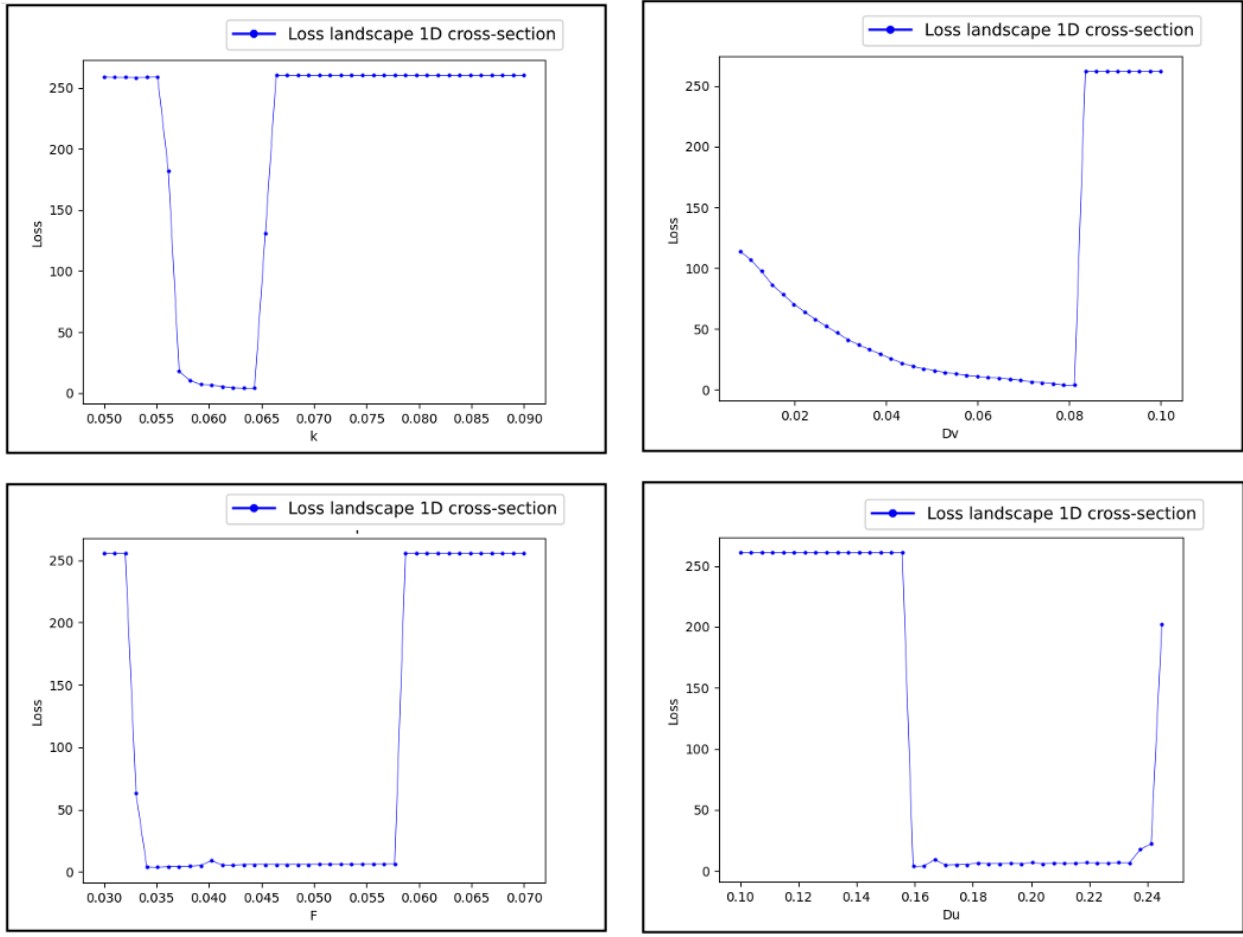

Figure 2: Loss landscapes cross-sections along single parameters. **Left-top:** $k$. **Left-bottom:** $F$. **Right-top:** $D_v$. **Right-bottom:** $D_u$. All other parameters are fixed at their respective ground truth values. The $D_v$ plot is missing a left high-loss region, and the $D_u$ plot missing a right one, both because $v$ values diverge to Inf at those ends, making loss computation infeasible.

The cross-sections reveal sharp cliffs separating distinct regions of the parameter space, with extensive flat areas on both sides providing negligible gradient signal. These features are consistent across all four one-dimensional axes and both two-dimensional cross-sections investigated.

### 4.2 Cross-Sections for Different Loss Functions

To verify whether the same issues occur across different loss functions, we compare three loss functions on the same cross-section of $F$-$k$. In Figure 3, the right plot already shows such a landscape for non-windowed 2D power spectrum loss, while Figures 5a and 5b provide corresponding three-dimensional plots for the non-windowed and windowed versions. These are qualitatively similar.

We also evaluate a third loss function, the VGG-based Gram matrix loss. Specifically, both the generated and target patterns are passed through the first 18 layers of a pretrained VGG-19 network (Simonyan & Zisserman, 2015), and the loss is computed as the mean squared error between their respective Gram matrices in feature space. This loss was originally developed for neural style transfer (Gatys et al., 2016), where Gram matrix similarity captures the statistical distribution of features rather than pixel-level correspondence. Unlike power spectrum losses, which measure frequency-domain similarity of the final patterns, the VGG

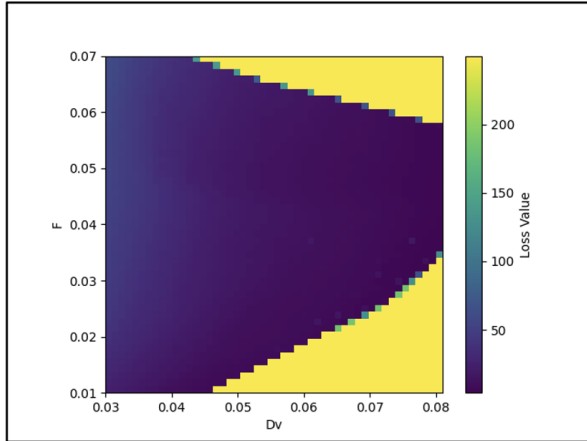 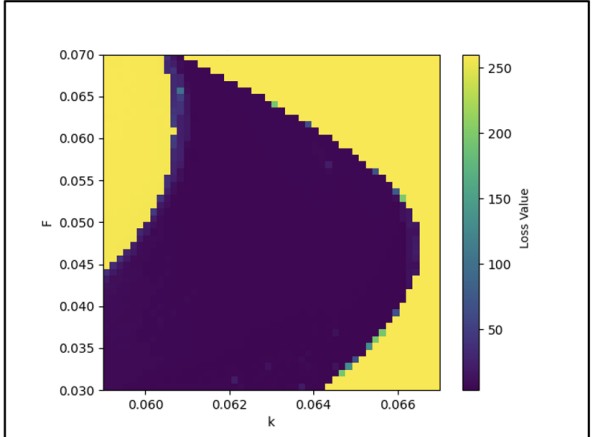

Figure 3: Loss landscape cross-sections on 2-dimensional planes formed by varying pairs of parameters, both of $50 \times 50$ granularity. Parameters not showing on a plot are fixed at their respective ground truth values. **Left:** $F$-$D_v$. **Right:** $F$-$k$.

Gram loss operates in a learned feature space sensitive to texture and structural regularity. We probe its loss landscape to assess whether a loss function that captures higher-level pattern statistics, rather than explicit frequency content, yields more navigable geometry.

As we can see in Figure 5c, the VGG loss differs from its power spectrum companions in three ways: 1) values span a smaller range than the power spectrum versions ($0 \sim 100$ versus $0 \sim 200+$); 2) the uniform-solution plateau is at a mid-level value, unlike the power spectrum losses where this plateau dominates at high values; and 3) the pattern-forming region exhibits loss fluctuations rather than remaining at low values as in the power spectrum cases.

Despite these differences, sharp discontinuities still dominate the boundary between the uniform-solution and pattern-forming regions, and the plateaus retain negligible gradient signals. The only navigable slope in the middle of the pattern-forming region is isolated by sharply rising ridges on one side and steep cliffs leading to higher plateaus on the other side. Illustrations from different angles in Appendix C (Figure 8) show this more clearly. The fluctuations in the pattern-forming region reveal the VGG loss's sensitivity to pattern differences. However, these fluctuations introduce additional cliffs within the pattern-forming region itself, making it equally unnavigable.

## 5 Discussions

### 5.1 Empirical Result Interpretation

All the loss landscape cross-sections we plotted for Section4 (Figures 2, 3, and 5) are dominated by plateaus with negligible signals and sharp cliffs. Typical cliffs separating pattern-forming with uniform-solution regions likely arise at bifurcations—the locations are surprisingly close to the bifurcations shown in Figure 2 of Delgado et al. (2017) and Figure 11 of Gandy & Nelson (2022). They form similar cusps, and small location deviations are likely due to different settings of diffusion coefficients. We speculate that, at saddle-node bifurcations, the pattern-forming solutions suddenly emerge, becoming the only or stronger attractor, while the uniform solutions dominate on the other side. The Hopf bifurcation is also a candidate here—limit loops observed in our $v$ animations often remain uniform for a long time with occasional flashes of dynamical local patterns, which makes it tempting for our step algorithm to cut them off as uniform solutions. Our optimizer was mostly confined to the uniform-solution (or quasi-uniform-solution limit-loop) regions, which explains the unchanging high losses observed during training. Its very occasional excursions into the pattern-forming

region remain brief: gradient signals there are weak and noisy, and the bounding cliffs, though steep, are narrow enough that a single misguided step crosses them, pushing the optimizer back into the plateau.

We reason that the gradient-based approach inherits this problematic landscape regardless of how gradients from the steady-state discrepancy are fed back to the parameters: whether via backpropagation through unrolled steps, implicit differentiation, or a forward surrogate that maps PDE parameters to steady-state patterns. Alternatives along these lines therefore do not change the nature of the problem.

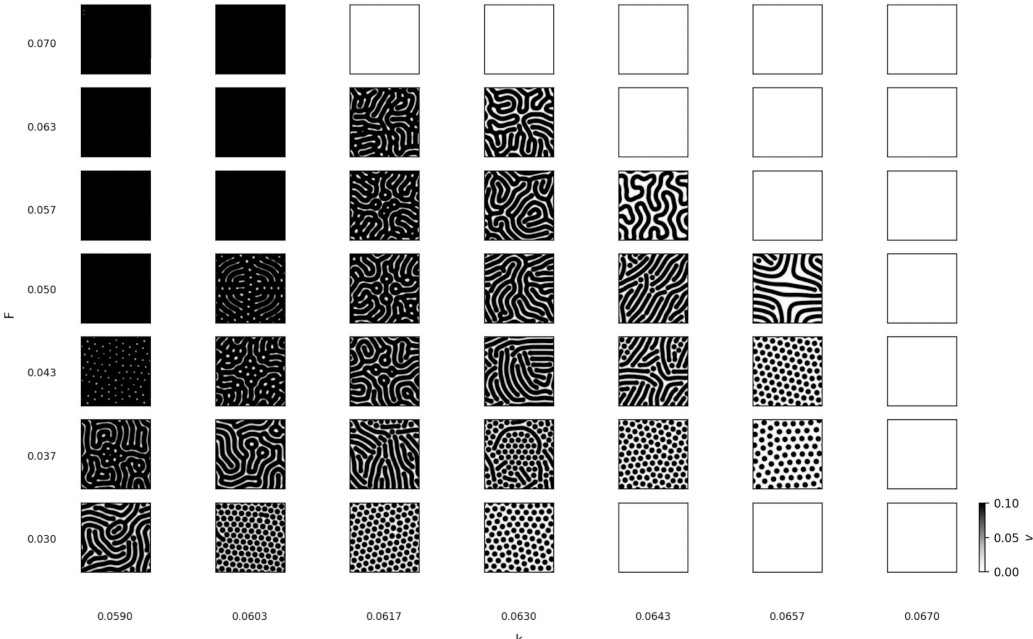

Figure 4: Steady-state patterns behind the $F$-$k$ loss plot (on the right part of Figure 3).[3]

Despite these challenges, the pattern matrix in Figure 4 reveals a potentially exploitable structure. On the $F$-$k$ plane, it shows gradual and regular pattern changes on the pattern-forming side of the cliffs, suggesting that this region is in principle traversable. We can similarly observe such gradual changes on the low-loss cliff sides in the cross-sections of Figure 2, although almost undiscernible in the cases of $F$ and $D_u$. This traversability, however, is not accessible from the uniform side due to the sharp cliffs, besides being a weak and unstable signal even when accessible.

We further observe, through evolution animations such as those linked in Appendix B and D, that patterns get more similar to one another as they evolve in time, and that uniform steady-states evolve from patterns in their earlier time steps.[4] Therefore, the regular changes like those in steady-state pattern regions should exhibit themselves even more prominently at intermediate time steps prior to convergence—and, at those earlier steps, regular pattern change may be present on both sides of the cliff, including in regions where patterns ultimately converge to uniform solutions. To what extent this can be exploited remains to be explored.

---

[3]Same colormap clipping as in Figure 1 (Footnote 2)

[4]We verified this with much more diverse initial conditions, including fully random initialization, confirming it is not an artifact of similarly initialized fields; additional figures and animations are available in Appendix E.

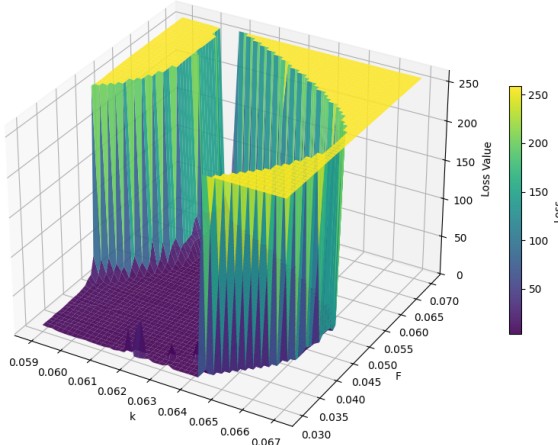

(a) Cross-section for the non-windowed 2D power spectrum loss.

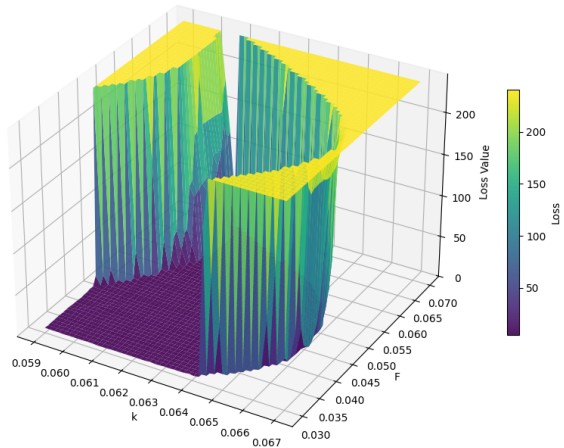

(b) Cross-section for the windowed 2D power spectrum loss.

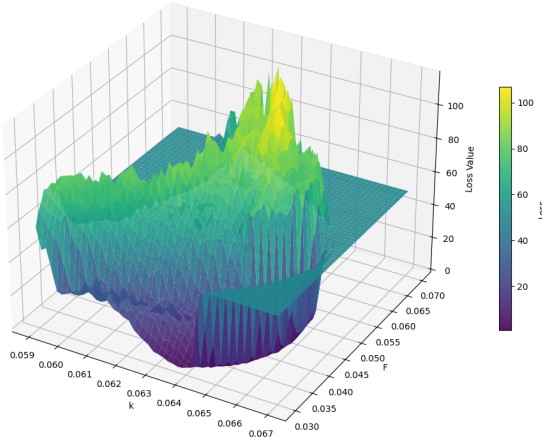

(c) Cross-section for the VGG-based Gram matrix loss.

Figure 5: 2-Dimensional Loss Landscape Cross-Sections for Different Loss Functions, all of $50 \times 50$ granularity. Parameters $D_u$ and $D_v$ are fixed at ground truth values.

### 5.2 Possible Remedies for the Landscape Pathologies

Continuing from the diagnosis so far, we briefly outline several possible directions to overcome the landscape issues, noting that the analyses in the following section will offer a different perspective. **(a) Better loss function design:** a loss function could provide meaningful gradient signal within the pattern-forming region while incorporating a mechanism to escape the uniform steady-state plateau. The regularity of pattern variations there suggests such a design is achievable. **(b) Time-augmented surrogate model:** a surrogate neural network trained to map $(D_u, D_v, F, k, t) \to pattern$, with simulation time $t$ as an extra input dimension, can enable the optimizer to explore earlier time steps where the parameter subspace exhibits more tractable loss geometry, and subsequently transfer progress toward improved final-state predictions. **(c) Intermediate supervision:** incorporating supervision at intermediate time steps may provide gradient signals unavailable from terminal steady-state comparisons alone. This requires careful design, as ground truth for intermediate states is not available in our setting.

## 6 Disentangling PINN

The above remedies all attempt to somehow reshape the problematic landscape. A prior question is whether existing methods already avoid it—which reframes our diagnostic probe as an ablation of PINN. Upon confirming that PINN does avoid the issue, this ablation leads us to disentangle which component is responsible, and in turn points to design implications for PINN-type methods and a more general heuristic for navigating pathological parameter landscapes beyond PDEs.

### 6.1 Existing Methods and the Case for PINN

Surveying the general literature on this type of inverse problem reveals limitations of non-PINN approaches in our setting. Schnörr & Schnörr (2023) implemented a surrogate model that maps patterns to PDE parameters in a direction opposite to the mapping with an ill-posed landscape, thereby avoiding landscape issues. However, their model only aims at coarse parameter estimation and is trained to conflate different initial-condition variations. The issue of different parameter sets generating very similar patterns can pull their surrogate toward conflicting training instances, causing it to learn a compromised representation. Najarro et al. (2026) address the related issue of noisy initial conditions—where the same parameters produce visually similar but pixel-level-different patterns—by using visual embedding distance as a loss function, a contribution consistent with our VGG-based loss. Their visual embedding loss shows discernibility across discrete parameter samples in the pattern-forming region, echoing our observation in Section 5.1 that pattern/loss changes do exist. However, discernibility at discrete points does not imply traversability across the full continuous landscape, and their method relies entirely on evolutionary search, which is computationally expensive and does not exploit loss gradients.

PINN naturally avoids both of these issues: it uses gradient-based optimization directly, and, without a reverse-mapping surrogate that conflates conflicting training instances, it searches on the original loss space and can converge to one of the multiple minima. However, literature suggests that applying standard or improved PINN frameworks to our type of inverse problems is underexplored. For example, attempts to apply PINNs to the Gray-Scott model in Giampaolo et al. (2022) addressed only forward problems and did not seek inverse parameter estimation. The degree of success reported in Zheng et al. (2024) covered only one specific set of Gray-Scott parameters at substantial computational cost. Most importantly, they did not provide a theory about how landscape issues were resolved. We therefore examine more carefully why PINN's advantages over non-PINN methods translate or fail to translate to our setting.

### 6.2 The Residual Loss Yields a Well-Behaved Landscape

We first adapt the PINN formulation to our setting and set up the joint-space geometry, then show that the residual loss alone yields a well-behaved landscape.

In the first PINN paper (Raissi et al., 2019), a neural network is trained with fixed PDE constraints to fit the data mapping $(x, y, t) \to (u_{x,y}, v_{x,y})$, which is equivalent to $(t) \to (U, V)$, where $U$ and $V$ represent full

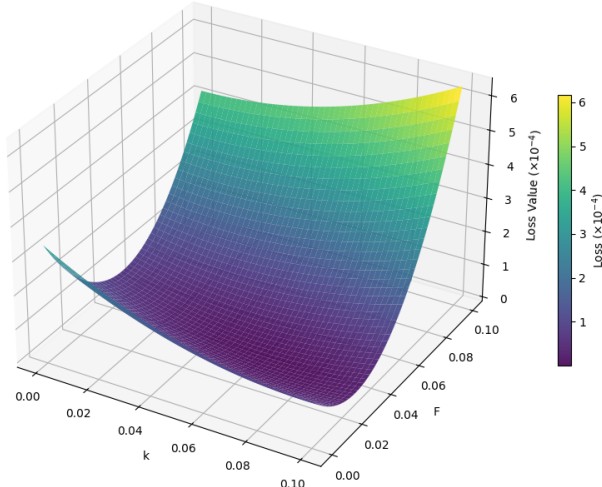

Figure 6: Residual loss cross-section formed by varying parameters $F$ and $k$, in $50 \times 50$ granularity. Parameters $D_u$ and $D_v$ are fixed at ground truth values.

images at time step ($t$). This equivalence holds because at any time step, the pair of full images equals a mapping from index pairs $(x, y)$ to pixel value pairs $(u_{x,y}, v_{x,y})$. The same PINN paper (Raissi et al., 2019) then made the PDE parameters learnable, training them together with the neural network parameters, while keeping the data mapping the same. To adapt it to our problem, we need to set all $t$'s to a large constant (to use steady-state patterns without time labels) and evaluate the residual loss as the squared residual of the elliptic form.

With this background, we consider the joint parameter space $(\theta, \mu) \in \mathbb{R}^n \times \mathbb{R}^m$, where $\theta$ denotes neural network parameters and $\mu$ denotes PDE parameters, to analyze its geometry. It is trivial that these two subspaces are orthogonal: for any $v \in \mathbb{R}^n$ and $w \in \mathbb{R}^m$, $(v, 0) \perp (0, w)$. The total loss decomposes as

$$L(\theta, \mu) = L_{\text{data}}(\theta) + L_{\text{res}}(\theta, \mu), \tag{7}$$

where the data term depends on $\theta$ alone.

To make comparisons with the losses plotted in Section 4, we study the loss restricted to the PDE parameter subspace, and, since $L_{\text{data}}$ is independent of $\mu$, we only consider $L_{\text{res}}(\theta, \cdot)$, residual loss for the $\mu$-subspace at fixed $\theta$. During the training of a PINN, a fixed $\theta$ produces an output somewhere between random initialization and the target, depending on how well the network has been fit at that moment. Because, theoretically, $L_{\text{res}}(\theta, \cdot)$ compares this fixed outcome of neural network with the solution of the PDE, one may expect to see the same ill-posed landscape. However, direct calculation shows that, once $\theta$ is fixed, the network outputs $u$, $v$ (and hence $\Delta u$, $\Delta v$, and $uv^2$) are all fixed, and the residual of the elliptic Gray-Scott equation is linear in parameters $\mu$. Therefore, the residual loss is a quadratic function of the parameters, yielding a smooth, bowl-shaped landscape. Our empirical cross-section in Figure 6 confirms this bowl shape.

This is an interesting counter-intuition: whatever the three directions in Section 5.2 endeavor to achieve, residual loss has already achieved neatly, with no contribution from the neural network. Looking at why, we find that, although residual loss compares the target with what the PDE actually produces, its comparison implicitly includes the full evolution dynamics, considering all initial conditions at once, rather than comparing the final pattern of a given IC. This allowed residual loss to access much more abundant information, including the intermediate steps we sought to utilize across the directions in Section 5.2.

### 6.3 A Neural Network Cannot Repair an Ill-Posed Landscape

A converse question is whether a neural network can *repair* a landscape that is already ill-posed: suppose a hypothetical system whose parameter space (denoted $\tilde{\mu}$) is itself ill-posed under a residual or alternative loss $\tilde{L}_{\text{res}}$—does adding the PINN-style auxiliary (a neural network $\theta$ and a data loss) help? The short answer is no.

In the $\theta$-subspace (for any fixed $\tilde{\mu}$), both the data loss and residual loss landscapes are typically amenable to gradient-based optimization, as is common when fitting a neural network to fixed data (Sitzmann et al., 2020). Although Krishnapriyan et al. (2021) characterize PINN failure modes in a setting different from ours—the forward problem of recovering the full space-time solution at large PDE coefficients—their analysis bears on the present point as the $\theta$-subspace we consider coincides with the network-parameter landscape they study. Their findings in fact support our assessment: their sequence-to-sequence remedy shows that fitting only a short slice of time with residual loss is tractable, and our steady state corresponds to a single such slice.

Yet a tractable $\theta$-subspace cannot rescue an ill-posed $\tilde{\mu}$-subspace, as the gradient of the total loss $L(\theta, \tilde{\mu}) = L_{\text{data}}(\theta) + \tilde{L}_{\text{res}}(\theta, \tilde{\mu})$ reveals. This gradient splits into three components:

$$\nabla L = \begin{pmatrix} \nabla_\theta L_{\text{data}} + \nabla_\theta \tilde{L}_{\text{res}} \\ \nabla_{\tilde{\mu}} \tilde{L}_{\text{res}} \end{pmatrix}, \tag{8}$$

and two reasons lead to this conclusion.

First, movement within the $\theta$-subspace at one particular point in the $\tilde{\mu}$-subspace does not transfer well to neighboring points in the $\tilde{\mu}$-subspace. This failure does not stem from $\nabla_\theta L_{\text{data}}$—$L_{\text{data}}$ is independent of $\tilde{\mu}$, and its $\theta$-landscape remains identical at different points in the $\tilde{\mu}$-subspace. The failure comes from $\nabla_\theta \tilde{L}_{\text{res}}$, where each $\tilde{\mu}$ defines a different target pattern for the neural network, and consequently a different loss landscape $\tilde{L}_{\text{res}}(\cdot, \tilde{\mu})$. If this target pattern jumps at discontinuities, slices of loss landscape $\tilde{L}_{\text{res}}(\cdot, \tilde{\mu})$ can change discontinuously due to jumps along $\tilde{\mu}$.

Second, no matter how the optimizer moves in the $\theta$-subspace, the harshness of landscapes $\tilde{L}_{\text{res}}(\theta, \cdot)$ persists. $\tilde{L}_{\text{res}}(\theta, \cdot)$ at any $\theta$ inherits the difficulties of $\tilde{\mu}$-subspace, making $\nabla_{\tilde{\mu}} \tilde{L}_{\text{res}}$ non-informative.

Thus, while PINN increases the dimensionality of the search space, this extra freedom does not provide an effective detour when we have a problematic landscape structure. The well-behaved $\theta$-subspace noted above is, moreover, the most favorable case for PINN: were either of these landscapes less well-behaved, this conclusion would only be reinforced. Whether the residual loss landscape over $\theta$ remains well-behaved beyond our single-frame setting—in particular in the full space-time problem, where the evidence of Sitzmann et al. (2020) and Krishnapriyan et al. (2021) appears to conflict—is left to future work.

### 6.4 Roles of Components and Design Implications

We now have the conclusions that residual loss alone has prevented the landscape issue, and that the neural network does not contribute more to this endeavor. It follows directly that the neural network serves only to complete the data.

Based on this theory of what each component of PINN does, we can optimize and reduce redundancy of PINN's neural network component, use the data scheme (what is observed and how much is observed) of the specific application to inform the design of the neural network, and adjust the combination of neural network and residual loss according to the training requirements of the specific application. We have already developed detailed designs for these directions but intend to disclose them in future publications rather than in this work.

The finding that a neural network cannot improve an ill-posed PDE parameter landscape further suggests a general design heuristic beyond the PDE setting: when loss landscapes are pathological in a given parameter subspace, any auxiliary dimensions introduced should provide navigable detours around the pathological structure, rather than leaving it insurmountable in the expanded space. The $t$-enhanced parameter space in direction (b) of Section 5.2 is one example of applying this principle.

# 7 Conclusion and Future Work

Serving as a radical ablation of the full PINN framework, we directly applied gradient-based optimization to recover Gray-Scott system parameters from IC-dependent steady-states without any auxiliary structure. Empirical analysis revealed severe pathologies in the resulting loss landscapes—sharp cliffs near bifurcation boundaries and flat plateaus in uniform-solution regions—that systematically prevent convergence.

Through principled theoretical analysis, further supported by experimental verification of the residual loss landscape, we then disentangled the roles of PINN's components: the residual loss avoids these pathologies by implicitly encoding full PDE dynamics, rather than single-trajectory steady-state information, yielding a smooth quadratic landscape in PDE parameter space; while the neural network, unable to improve this landscape, serves instead to complete partial observational data—a distinction not previously made explicit.

The next steps of our work follow the design implications in Section 6.4: testing neural network architectures tailored to the data completion role identified here, optimized for different observational schemes across diverse PDE systems and other physical domains. We have already completed a redesign for the Gray-Scott inverse problem discussed in this paper and intend to disclose the details in future publications.

Two further directions concern the analysis itself. First, the good behavior of the residual loss landscape over $\theta$—argued here by analogy to Sitzmann et al. (2020) and Krishnapriyan et al. (2021)—warrants direct empirical verification, through landscape visualization, in our own steady-state Gray-Scott setting. Second, we aim to resolve the open question raised in Section 6.3: whether this landscape remains tractable in the full space-time setting, and what underlies the conflicting evidence noted there, through theoretical analysis and further visualization.

## Software and Data

All code and experimental data to reproduce our results are publicly available in an anonymous GitHub repository: `https://github.com/TMLR-submission/bp_inversion`.

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

## A    Boundary and initial conditions

The stepping algorithm uses the periodic boundary condition. Initial conditions are generated randomly for each training step, following the same controlled pattern:

$$u_{i,j} = \begin{cases} 0.50, & \text{if } 54 \le i, j \le 74 \\ \text{clamp}(1.0 + \mathcal{N}(0,1), [0.0, 1.5]), & \text{otherwise} \end{cases} \tag{9}$$

$$v_{i,j} = \begin{cases} 0.25, & \text{if } 54 \le i, j \le 74 \\ \text{clamp}(\mathcal{N}(0,1), [0.0, 1.0]), & \text{otherwise} \end{cases} \tag{10}$$

.

## B    The One Training Trajectory Manually Picked

Here we show a trajectory that the optimizer followed but did not automatically pivot or cut off. The pivoting and cut-off points are identified by manually animating pattern generation at random checkpoints and comparing them with targets. The trajectory record:

- Initial parameters $D_u = 0.1270$, $D_v = 0.1269$, $F = 0.0500$, $k = 0.0501$.

- Initial training: used L2 loss on 2d power spectrum, learning rate $1.2e - 2$, and trained for 8918 iterations, arrived at $D_u = 0.1285$, $D_v = 0.0734$, $F = 0.0429$, $k = 0.0682$.

- Continued training using L2 loss on 2d power spectrum and learning rate $1e - 3$, for 98 iterations, arrived at $D_u = 0.1343$, $D_v = 0.0679$, $F = 0.0429$, $k = 0.0653$.

We also created a gif animation file, showing simultaneous animations of four pattern evolutions, under: 1) initial parameters, 2) parameters after the initial 8918 iterations, 3) parameters after the extra 98 iterations, and 4) target parameters. The four animations are arranged from left to right in the file: https://osf.io/xfk7z/files/348kw?view_only=e26ec2a6706c40a4a0606691f9449900.

## C   More Cross-Section Figures

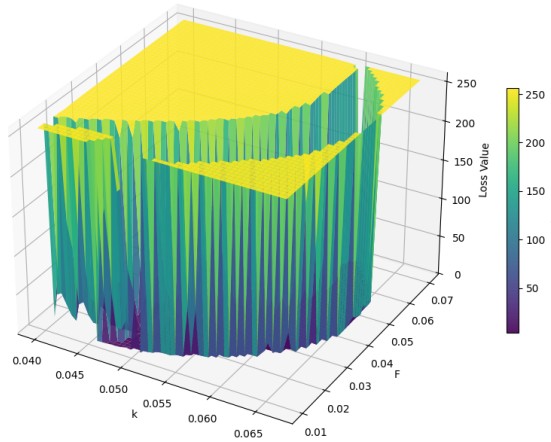

Figure 7: Non-windowed loss landscape on a larger region of $F$-$k$, $30 \times 30$. This shows that the lower plateau is surrounded by cliffs from all sides.

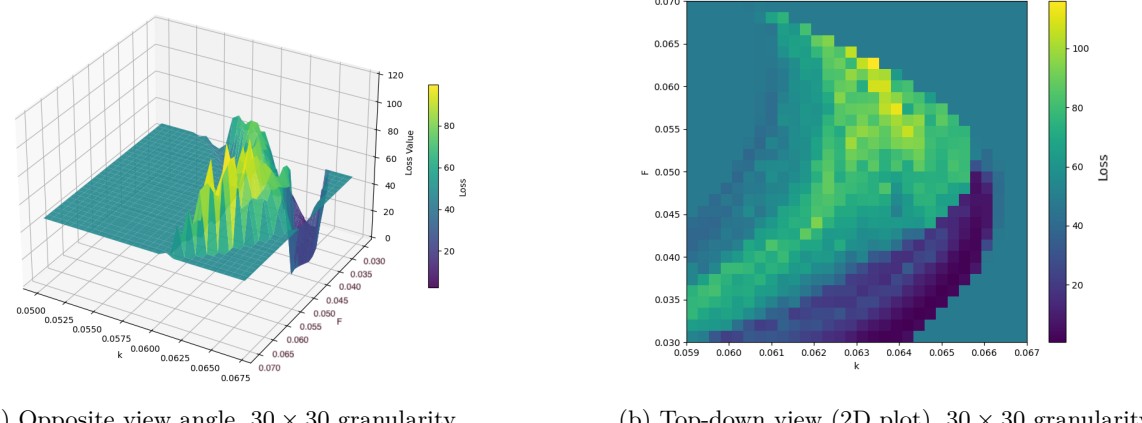

(a) Opposite view angle, $30 \times 30$ granularity.   (b) Top-down view (2D plot), $30 \times 30$ granularity.

Figure 8: Extra $F$-$k$ cross-section plots for the VGG-based Gram matrix loss landscape.

## D   Links to Animation Files

Each file shows 12 simultaneous animations, corresponding to 12 sample values of the investigated parameter, with the other 3 parameters set to their ground truths.

- Along $k$: `https://osf.io/sgf52?view_only=e26ec2a6706c40a4a0606691f9449900`

- Along $F$: `https://osf.io/423hm?view_only=e26ec2a6706c40a4a0606691f9449900`

- Along $D_u$: `https://osf.io/xfk7z/files/mc582?view_only=e26ec2a6706c40a4a0606691f9449900`

- Along $D_v$: `https://osf.io/xfk7z/files/gzujm?view_only=e26ec2a6706c40a4a0606691f9449900`

## E   Intermediate vs Steady-State Patterns Under Varied Initial Conditions

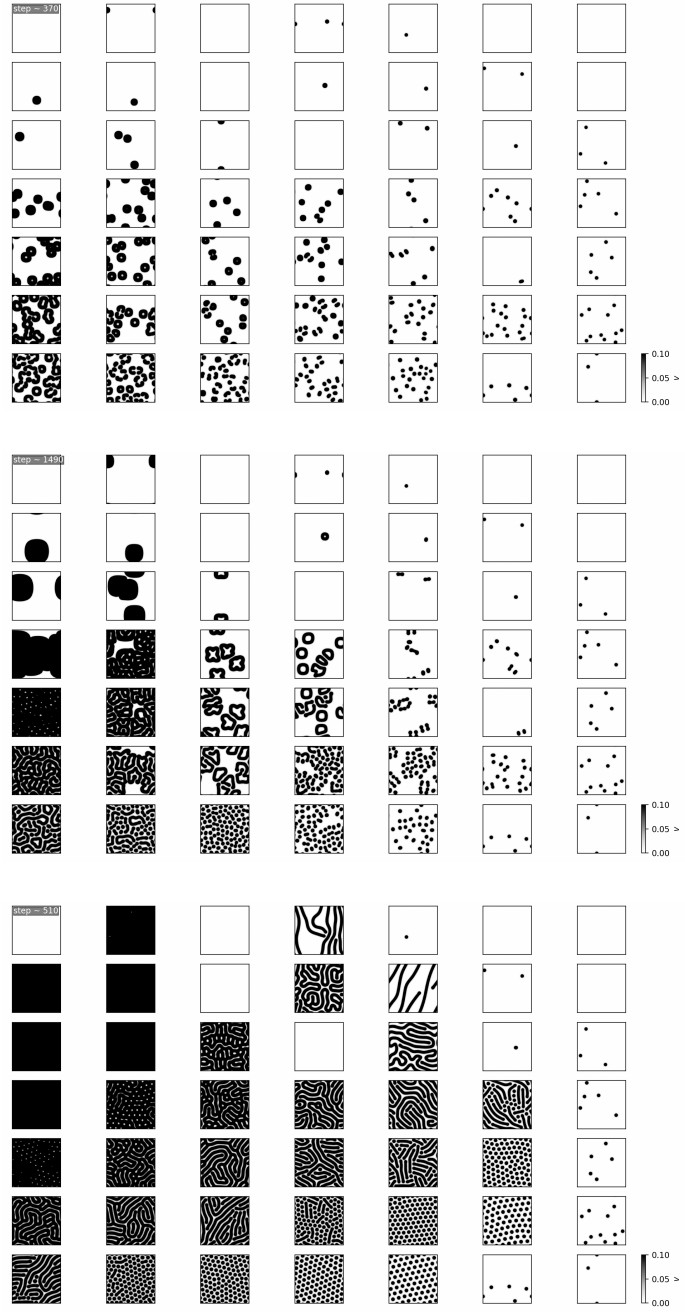

Figure 9: Intermediate (top two) and steady-state (bottom) patterns for large (6× the original) random noise.

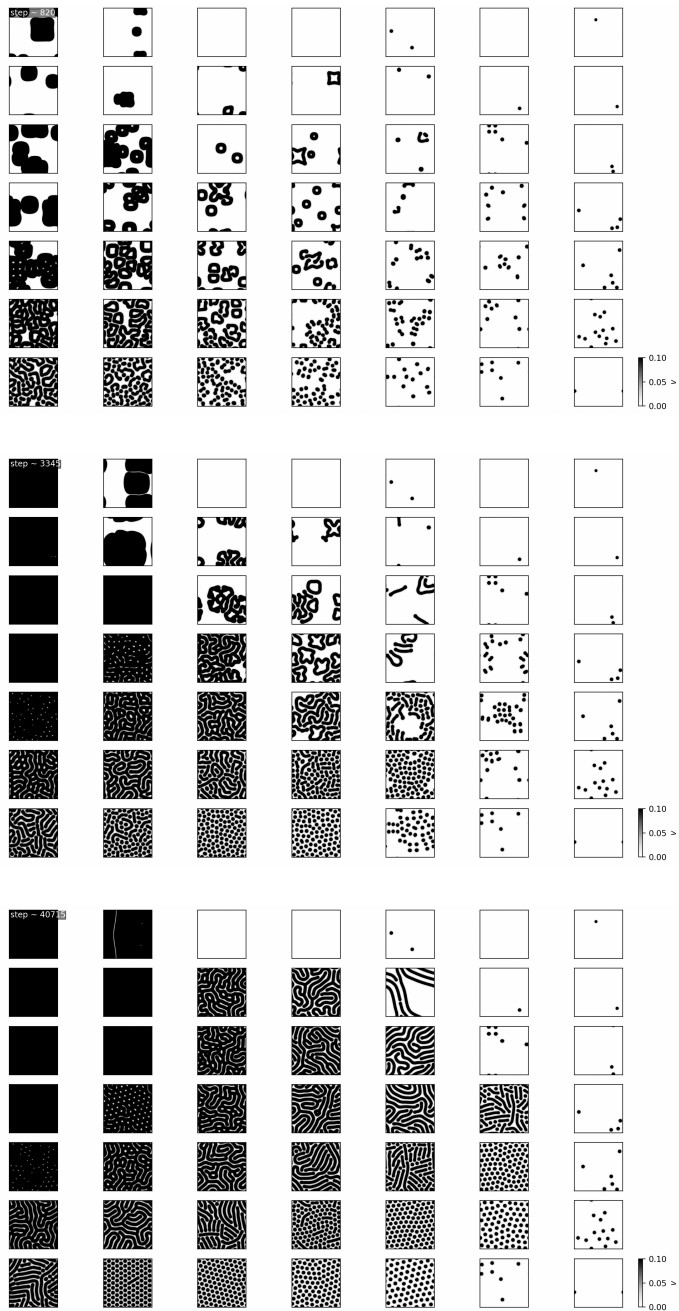

Figure 10: Intermediate (top two) and steady-state (bottom) patterns for random noise with a random-sized perturbation box at a random location.

