# OpenReview forum: "Loss Landscape Diagnosis for Gradient-Based Gray-Scott System Inversion: Disentangling the Roles of PINN Components"
_TMLR — Under review for TMLR_

### Review · Reviewer_QLQz · 2026-06-15

**Summary Of Contributions:**

This paper investigates why direct gradient-based inversion of the Gray-Scott reaction-diffusion system fails. The authors backpropagate a steady-state loss through unrolled simulation steps (without neural network augmentation or surrogates) to recover four PDE parameters, and document systematic failure. Through landscape visualization, they reveal pathological geometry: flat plateaus with negligible gradient signal bounded by sharp cliffs that align with bifurcation boundaries of the PDE. This structure persists across three loss functions (2D power spectrum, windowed power spectrum, VGG Gram matrix loss).

Framing this minimal setup as an ablation of Physics-Informed Neural Networks (PINNs), the authors disentangle each component's role: (1) with the neural network output fixed, the residual loss is quadratic in PDE parameters, yielding a smooth bowl-shaped landscape; (2) the neural network cannot repair an ill-posed parameter subspace and serves only to complete partial observations. This division of labor is stated as not previously made explicit.

**Key Strengths:**
- The conceptual framing of direct backpropagation as a "radical ablation of PINN" is creative and yields a genuinely novel decomposition of component roles.
- The connection between landscape pathology and PDE bifurcation structure is scientifically grounded and informative.
- Landscape visualizations (Figures 2-5) are thorough and clearly communicate the geometry of the problem.
- Code is publicly available for reproducibility.

**Key Weaknesses:**
- Only one PDE system (Gray-Scott) is studied, with no evidence that conclusions generalize.
- No actual PINN is run to empirically validate the theoretical claims about residual loss avoiding pathology.
- The paper is primarily diagnostic with no constructive resolution; solutions are explicitly deferred to "future publications."
- Quantitative metrics of failure/success are absent; the analysis remains largely qualitative.

**Additional Comments:**

This paper contains a genuinely interesting core insight: the explicit decomposition of PINN into "residual loss handles landscape geometry" and "neural network handles data completion" is novel and, if properly validated, would be a useful contribution to the scientific ML community. The landscape visualizations are thorough and the writing is generally clear.

However, in its current form, the paper reads as the first half of a complete story. The diagnostic phase is well-executed, but the validation of the key theoretical conclusions (Section 6) requires empirical support that is currently absent. The most impactful single addition would be running an actual PINN on this problem and demonstrating success, thereby closing the loop between "direct approach fails" and "PINN avoids this failure via the residual loss mechanism identified here."

I also note that the observation in Section 6.2 that residual loss is quadratic in PDE parameters (given fixed network output) follows relatively straightforwardly from the fact that the Gray-Scott equations are linear in $(D_u, D_v, F, k)$. While the authors correctly identify this, it would be valuable to discuss what happens for PDEs where parameters enter nonlinearly, as this would clarify the scope of the insight.

The paper is well-positioned for acceptance after revision if the critical changes above (particularly items 1-3) are addressed. The underlying ideas are sound and interesting, but the current evidential standard falls short of what is needed to support the breadth of the claims made.

**Audience:**

Yes

**Audience Explanation:**

The paper addresses a question of genuine interest to the scientific machine learning community: why do PINNs work for PDE inverse problems, and what role does each component play? The specific findings would interest several audiences:

1. **PINN practitioners** would benefit from the explicit decomposition of component roles (residual loss handles landscape geometry; neural network handles data completion). This has direct design implications, as noted in Section 6.4.

2. **Researchers studying PDE inverse problems** would find the landscape pathology analysis informative, particularly the connection between optimization difficulty and PDE bifurcation structure.

3. **Dynamical systems researchers** may find the empirical evidence of how bifurcation boundaries manifest as optimization barriers useful when designing inversion algorithms for pattern-forming systems.

4. **The broader ML optimization community** may find the general heuristic (Section 6.4) relevant: "when loss landscapes are pathological in a given parameter subspace, any auxiliary dimensions introduced should provide navigable detours around the pathological structure."

The topic is timely given the growing interest in PINNs and neural PDE solvers, and the negative result (direct backpropagation fails, here is why) is itself informative. The paper would benefit a segment of TMLR's audience even in its current form, though the impact would be substantially greater with the additional experiments described below.

**Broader Impact Concerns:**

No concerns. This is a diagnostic analysis of optimization landscapes for a mathematical model of chemical pattern formation. It poses no ethical risks, raises no dual-use concerns, and does not involve human subjects, sensitive data, or potentially harmful applications. The work is purely methodological in nature.

**Claims And Evidence:**

No

**Claims Explanation:**

The paper makes two central claims: (1) direct gradient-based inversion fails due to landscape pathology, and (2) PINN avoids this pathology because the residual loss yields a well-behaved quadratic landscape while the neural network only completes data. While claim (1) is well-supported through extensive landscape visualizations and experiments with multiple loss functions, claim (2) has significant evidential gaps:

1. **No empirical PINN validation.** Section 6.2 argues theoretically that the residual loss landscape is quadratic in PDE parameters and provides one empirical cross-section (Figure 6). However, no actual PINN is run on the Gray-Scott inverse problem to demonstrate that it succeeds where direct backpropagation fails. This is the most natural and critical experiment to validate the paper's central theoretical conclusion, and its absence is a major gap.

2. **Single cross-section for the key claim.** The "well-behaved" residual loss landscape is shown via one 50x50 cross-section in the $F$-$k$ plane (Figure 6). It is not demonstrated across different neural network initializations, architectures, or training stages. The quadratic argument assumes fixed network outputs, but during actual PINN training these outputs change continuously.

3. **Informal reasoning for Section 6.3.** The claim that "a neural network cannot repair an ill-posed landscape" rests on an informal "two reasons" argument rather than a rigorous proof or systematic empirical demonstration. The gradient decomposition in Eq. (8) is correct but the conclusion drawn from it (that movement in the $\theta$-subspace cannot help) is argued qualitatively.

4. **Generality claimed but not demonstrated.** The paper states it aims "for conclusions that extend to broader reaction-diffusion and PDE inverse problems" (Section 1), but all evidence comes from a single PDE system. The authors speculate about connections to bifurcation theory (saddle-node, Hopf) but do not verify these connections rigorously.

5. **Qualitative failure characterization.** The failure of direct backpropagation is documented via the observation that "loss stays within a narrow high band of 245.0-270.0" (Section 3.1) and one manually-picked trajectory (Appendix B). No systematic metric (e.g., success rate over random initializations, distance of final parameters from ground truth across runs) is provided.

The landscape visualizations supporting claim (1) are convincing, and the theoretical argument for claim (2) is plausible. However, the standard for "accurate, convincing and clear evidence" requires either rigorous proof or empirical demonstration of the key conclusions, and neither is fully provided for the PINN-related claims.

**Requested Changes:**

### Critical changes (required for acceptance):

1. **Run an actual PINN on the Gray-Scott inverse problem.** Implement a standard PINN (as formulated in Section 6.1-6.2) and demonstrate empirically that it recovers the Gray-Scott parameters where direct backpropagation fails. This is the most natural validation of the paper's central theoretical claim and its absence is difficult to justify given that the infrastructure for this experiment largely already exists. Show the residual loss landscape at multiple stages of PINN training to confirm the bowl shape persists dynamically.

2. **Test at least one additional reaction-diffusion system.** The paper claims generality ("conclusions that extend to broader reaction-diffusion and PDE inverse problems") based on a single system. Running the landscape analysis on even one additional system (e.g., FitzHugh-Nagumo, Schnakenberg, or Brusselator) would establish that the findings are not artifacts of Gray-Scott's specific structure. If this is infeasible, the generality claims must be substantially weakened.

3. **Provide quantitative failure metrics.** Replace or supplement the qualitative failure documentation with systematic measurements: (a) success rate of convergence over N random initializations (e.g., N=100), (b) distribution of final parameter distances from ground truth, (c) fraction of runs that enter the pattern-forming region of parameter space. This would transform the anecdotal evidence in Section 3.1 and Appendix B into rigorous characterization.

### Strongly recommended changes (would significantly improve the paper):

4. **Formalize the argument in Section 6.3.** The claim that neural networks cannot repair ill-posed landscapes is currently argued informally. Either provide a formal proposition with conditions under which this holds, or provide a concrete empirical demonstration (e.g., train a PINN on a system with genuinely ill-posed residual loss and show it fails in the $\tilde{\mu}$-subspace).

5. **Remove or substantiate the reference to "future publications."** The statement "we have already developed detailed designs for these directions but intend to disclose them in future publications rather than in this work" (Section 6.4) is inappropriate for a self-contained submission. Either include preliminary results in an appendix, or remove the reference entirely and let the diagnostic contribution stand on its own.

6. **Add comparison with at least one non-gradient-based inverse method.** Testing whether Bayesian inference, ensemble Kalman inversion, or even random search can solve this inverse problem would contextualize the difficulty. If these also fail, the finding is about the problem's intrinsic difficulty; if they succeed, it is specifically about gradient-based landscape navigation.

7. **Characterize the landscape resolution sensitivity.** The 50x50 grid for 2D cross-sections may miss fine-scale structure. Provide at least one higher-resolution cross-section (e.g., 200x200) in a region of interest to verify that the observed flatness is genuine rather than an artifact of coarse sampling.

### Minor improvements:

8. Add an explicit "Limitations" section acknowledging the single-system scope and informal nature of the Section 6.3 argument.

9. In Section 3.1, replace "we expect it would have" (speculation about the incidentally-found configuration) with either evidence or explicit acknowledgment that this is unverified.

10. Consider reporting the wall-clock time and GPU resources used for landscape evaluations (each 50x50 grid requires 2500 forward simulations to steady state), as this affects reproducibility.

---

### Review · Reviewer_m6Aj · 2026-06-15

**Summary Of Contributions:**

The paper solves the following problem. For single Gray-Scott equation we setup parametric solver and try, given observation data, to find Gray-Scott equation  parameters using gradient methods.

Gradient method fails and author try to find reasons using loss landscape visualization against multilple losses. They then retrospectively reframe this failed experiment as an ablation of PINN, using it to argue theoretically that the residual loss handles the landscape pathology while the neural network serves only data completion. No working method is demonstrated.

**Audience:**

Yes

**Audience Explanation:**

There are, in fact, three communities that interested in, let's say, "inverse problems": (I) equation discovery one, (II) PINN and (III) differentiable solvers. If we think if anyone needs such analysis, we move to:

(I) equation discovery

That is actually the paper tries to do. We have SINDy-like search, but the library is only all that Gray-Scott equation has.The library is not chosen by the method, it's fixed by assumption. That's actually a stronger limitation than SINDy, not a variant of it. Modification is that we replace direct derivatives computation with pre-defined solver. Within equation discovery the problem has merit, since in the paper authors look for the data with non-differentiable patterns.

For general case, equation discovery domain already has PIC [1], which is a more general thing and doesn't quite involve predefined solver. Also and more importantly it resembles the paper approach.

[1] Xu, H., Zeng, J., & Zhang, D. (2023). Discovery of partial differential equations from highly noisy and sparse data with physics-informed information criterion. Research, 6, 0147.

Additionaly, data approximation with neural networks is no new in equation discovery and serves to reduce noise before derivatives computation.

(II) PINN

Strictly speaking, there is no PINN and the paper and the authors agree with that. The idea to separate data loss and residual loss is harmfull and that is why:

PINN's residual loss doesn't just help with parameter recovery, it actively regularizes the neural network toward the solution manifold of the PDE. The network is constrained to live in a Sobolev space (with controlled derivatives), not just fit the data in an L2 sense. This is a feature, not incidental.

Neural network for data completion alone,so it will be regularized only toward data fit, so you're in something like W^{0,2} or just C^0 at best. To recover coefficients on that completed data you need accurate derivatives, but your network wasn't trained to have them.

(III) differentiable solvers

Is not covered at all, however, in my opinion, it is the best fit community for this paper.

**Broader Impact Concerns:**

I think that there could not be eithical concerns in PDE solvers.

**Claims And Evidence:**

No

**Claims Explanation:**

The paper revolves around single PDE, Gray-Scott equation and essentially tries to say that between its coefficients and different losses that reflect the data discrepancy (in general sense) there is hard to chose globally smooth function that has a minimum around the pattern forming modes. It is basically specific to a Gray-Scott spectral problem since such modes are point-wise spectra part, not the continious one.

There is a direct connection. There could not be the continious spectral problem for point-wise specter one, which reflects the authors search of one trough different loss functions.

Rare PDEs have point-wise spectra, thus, the evidences and claims could be valid for: (a) specific to Gray-Scott, (b) specific to reaction-diffusion systems near bifurcations, (c) general systems that allow for point-wise specter, (d) generic to nonlinear PDEs, (e) generic to any PDE inverse problem. Whereas (d) and (e) are unlikely, the paper's claims make impression that the authors state (e). However, based on a single equation one cannot distinguish between these cases.

In light of PINNs we have also rather general claim that "residual loss handles landscape pathology while neural network serves only data completion", not as a finding specific to Gray-Scott or reaction-diffusion systems.

**Requested Changes:**

1. I advise to find a "mental home" for the paper either (I), (II) or (III) (please refer to audience part) and state it explicitly.
2. Within the domain look at the real problem and also state it explicitly. For example: "we cannot discover equation for point-wise spectra problems and here is our explanation why" or "we cannot solve inverse problems with PINNs for point-wise spectra and here is our explanation why" (please, don't break PINNs anymore though) or "there are problems with differentiable solvers for certain cases and that's why"
3. Reduce claims to (a)-(c) (please refer to evidence section) depending on what are your abilities to do experiments and what they evident to.

---

### Review · Reviewer_LwZc · 2026-06-27

**Summary Of Contributions:**

The authors investigate the problem of recovering the ground-truth parameters of a Gray-Scott reaction-diffusion system from observed dynamics via backpropagation. The authors establish that the loss landscape of this problem is generally flat, with sharp cliffs close to the bifurcations. The extremely nonlinear landscape that is flat around the solution, with sharp cliffs around it makes it difficult for gradient descent to reach the solution. The authors show that three different loses give similarly nonlinear landscapes. How can we alleviate this pathology? The authors suggest 1) better loss design, 2) time-augmentation to the original problem, 3) supervision using additional signal other than the observed steady-state. More generally, the authors observe that PINNs are able to approach the solution more successfully compared to the direct backpropagation method that the authors have explored. The authors thus attempt to dissect how PINNs are able to do this in the second half of the paper. The authors note that once we introduce PINNs to the problem, the loss decomposes into components (which the authors call “data loss” and “residual loss”) where the residual loss is a quadratic function of the parameters of the Gray-Scott system, fixing the neural network parameters. The authors argue that PINNs may be more successful partly due to having achieved 1)-3). The authors then demonstrate that if the residual loss itself is not well-behaved like the quadratic function in this Gray-Scott example, but more convoluted, then neural networks themselves can’t make the residual loss more well-behaved.

**Audience:**

Yes

**Audience Explanation:**

Individuals who are interested in the Gray-Scott reaction-diffusion system, and how to recover the parameters of this system in a data-driven way may find this useful. More generally, individuals who work in data-driven dynamical models for scientific applications would be interested.

**Broader Impact Concerns:**

I don't see ethical concerns in this work.

**Claims And Evidence:**

Yes

**Claims Explanation:**

Partially yes.

In the first half of the paper, the authors demonstrate with multiple experiments and visualizations that flat surfaces/cliffs in the loss landscape of the Gray-Scott system exist and that they may be the sources of the pathology in training. Figures 1-5 are devoted to demonstrating this result.

The second half of the paper on how PINNs partly avoid the pathological landscape has relatively weak grounding, with Figure 6 being the only visualization for this section, demonstrating that the residual loss with respect to the system parameters is quadratic, when the neural network parameters are fixed. The authors suspect that this well-behaved quadratic loss is what contributes to the success of PINNs, and that if this loss was not well-behaved (e.g., having cliffs as in the first half), then PINNs will fail. The authors claim that the neural network serves only to complete the data.

While the authors give intuition into why PINNs may not be helpful in making the residual loss landscape well-behaved, their reasoning is limited to analyzing the loss with either the network parameters or the system parameters fixed, and to the well-behaved \theta-subspace case in the single-frame setting. The authors mention this limitation in Section 6.3 and in the last paragraph of the Conclusion.

The paper progresses by establishing that direct backpropagation method for recovering the parameters of Gray-Scott systems from data may have been avoided due to the pathological loss landscape, and then suggests some reasons for how PINNs may be alleviating this problem, and under what conditions PINNs may not be able to solve the problem. Based on this intuition, the authors mention that a future publication will show PINN designs that take into account their findings. I think that including the PINN designs in this work may make sense, given how the designs are directly motivated by the findings here. This could also serve to strengthen the second half of the paper, which has no experimental validation of the authors' claims. The second half, as it is, feels rather like a work in progress.

**Requested Changes:**

- In the first half of the paper, the authors' arguments rely only on the set of parameters: D_u = 0.16, D_v = 0.08, F = 0.035 and k = 0.065. Have the authors tried other sets of parameters for the ground truth? Showing similar results for a few other sets of parameters would strengthen the authors' claims.

- Observed data might contain noise, which the authors do not discuss. Does the paper assume a noiseless case, and does having noisy observations affect results in the paper?

- In Figure 1, the legend would be more clear if the authors stated that #2 is after 8918 iterations, and #3 is after 98 further iterations rather than combining their explanations as it is currently.

- The authors claim that loss values carry no convergence signal in their experiments. My reading of the paper was that the authors have always used 1e-2 as the initial learning rate (which adaptively changed over time). What if the learning rate was much lower? The authors rule out "insufficient exploration" on the basis of changing the loss function, but more hyperparameter search would strengthen their claim. Relatedly, I think that introducing the VGG-based Gram matrix loss along with the other two losses, rather than later on in the paper, would improve the clarity of the paper.

- The authors claim that neural networks in PINNs "serve only to complete the observed data". Could the authors elaborate more on why this is the case? I think this claim is related to the authors' argument in Section 6.3, but I don't fully understand. The authors mention that these findings carry design implications, but do not demonstrate an example of a new concrete design, and do not show empirically when the added dimension helps and when it doesn't. If the resubmission demonstrates a new design and more empirical results with different examples (not necessarily restricted to Gray-Scott system) on when augmenting additional variables to the network helps or not, this paper would be much stronger. This point is related to my comment above that the second half of the paper is rather weak compared to the first half.

- Generally, I found that references cited in the paper are rather sparse --  for example, the first couple of sentences in the introduction start with "Inverse parameter estimation that recovers the governing parameters of a dynamical system from observed outputs arises across scientific domains, from developmental biology (Kondo, 2022) to computational neuroscience (Lefèvre & Mangin, 2010). Reaction-diffusion systems are a canonical class of such problems: their parameters determine qualitatively distinct pattern-forming behaviors, and inferring them from steady-state or non-terminal observations has direct relevance to both physical modeling and biological pattern analysis (Kondo, 2022)." Please add more relevant references for developmental biology and computational neuroscience in this example, and generally throughout the paper.

- Relatedly, please give the relevant references for “Direct backpropagation is the most fundamental optimization mechanism in machine learning”. Is there supporting evidence that this is the most fundamental mechanism in the literature? The authors might consider changing the wording to something along the lines of backpropagation is widely used in deep learning to confine the scope of the claim.